# More Rule than Exception: Parallel Evidence of Ancient Migrations in Grammars and Genomes of Finno-Ugric Speakers

**DOI:** 10.3390/genes11121491

**Published:** 2020-12-11

**Authors:** Patrícia Santos, Gloria Gonzàlez-Fortes, Emiliano Trucchi, Andrea Ceolin, Guido Cordoni, Cristina Guardiano, Giuseppe Longobardi, Guido Barbujani

**Affiliations:** 1CNRS, UMR 5199—PACEA, Université de Bordeaux, Bâtiment B8, Allée Geoffroy Saint Hilaire, 33615 Pessac, France; sntprc1@unife.it; 2Dipartimento di Scienze della Vita e Biotecnologie, Università di Ferrara, 44121 Ferrara, Italy; gloria.gonzalezfortes@unife.it; 3Department of Life and Environmental Sciences, Marche Polytechnic University, 60131 Ancona, Italy; e.trucchi@univpm.it; 4Dipartimento di Comunicazione ed Economia, Università di Modena e Reggio Emilia, 42121 Reggio Emilia, Italy; ceolin@unimore.it (A.C.); cristina.guardiano@unimore.it (C.G.); 5School of Veterinary Medicine, University of Surrey, Guildford GU2 7AL, UK; g.cordoni@surrey.ac.uk; 6Department of Language and Linguistic Science, University of York, York YO10 5DD, UK; giuseppe.longobardi@york.ac.uk

**Keywords:** genomes, syntax, genetic and linguistic distances, human migrations, phylogenies

## Abstract

To reconstruct aspects of human demographic history, linguistics and genetics complement each other, reciprocally suggesting testable hypotheses on population relationships and interactions. Relying on a linguistic comparative method based on syntactic data, here we focus on the non-straightforward relation of genes and languages among Finno-Ugric (FU) speakers, in comparison to their Indo-European (IE) and Altaic (AL) neighbors. Syntactic analysis, in agreement with the indications of more traditional linguistic levels, supports at least three distinct clusters, corresponding to these three Eurasian families; yet, the outliers of the FU group show linguistic convergence with their geographical neighbors. By analyzing genome-wide data in both ancient and contemporary populations, we uncovered remarkably matching patterns, with north-western FU speakers linguistically and genetically closer in parallel degrees to their IE-speaking neighbors, and eastern FU speakers to AL speakers. Therefore, our analysis indicates that plausible cross-family linguistic interference effects were accompanied, and possibly caused, by recognizable demographic processes. In particular, based on the comparison of modern and ancient genomes, our study identified the Pontic-Caspian steppes as the possible origin of the demographic processes that led to the expansion of FU languages into Europe.

## 1. Introduction

Darwin proposed that linguistic diversity along human history tends to be correlated with the biological differentiation of populations [1]. Indeed, factors isolating populations from each other, such as geographical distance and barriers to migration, are likely to promote both biological and cultural divergence, whereas factors favoring contacts have the opposite effect at both levels [2,3,4,5]. In fact, despite élite dominance and other processes of horizontal language transmission creating local mismatches [6], parallel genetic and linguistic changes have often appeared as the rule rather than the exception [2,4,7,8,9,10]. This implies that linguistic diversity may offer a set of testable hypotheses about the demographic processes shaping genetic diversity, and vice versa.

In this study, through a multidisciplinary approach comparing grammars and genomes, we contribute to a better understanding of population diversity, both cultural and biological, in western/Central Eurasia. We focus on Altaic- (hereafter: AL) Finno-Ugric- (FU) and Indo-European- (IE) speaking populations, with a special emphasis on FU speakers. The reason is that FU has appeared as a possible exception to the general gene-language correspondence. Indeed, its monophyletic unity was acknowledged linguistically already in the 18th century [11] and (unlike AL, a linguistically controversial unit: see [12] for a summary) remains virtually undisputed (with the possible caveats in [13]), but FU-speaking populations fail to be identified as a genetic group [14]. In particular, the westernmost FU-speaking populations in northern and especially Central Europe have been shown to display a peculiar exception to the conclusion that, in Europe, grammars are better predictors of genetic distances than geography [4]. This exception is worth some further investigation.

Until recently, comparative studies of genes and languages suffered from serious limitations, simply because of the data available. On the one hand, only seldom were whole genomes considered in these comparisons. On the other, classical etymological comparison of vocabulary items, still normally used to reconstruct phylogenetic history even in modern quantitative studies (see, e.g., [15,16,17]), work well within a language family, but words cannot be used for broader comparisons: for, by definition, across different language families there are no recognizable common etymologies (i.e., lexical cognates; see Ref. [18] for an important attempt to remedy some of these problems). However, the Parametric Comparison Method (PCM) [19,20,21,22,23,24]), which explores the phylogenetic information contained in the generative rules of syntax, has in principle overcome the limitations of vocabulary-based taxonomic methods: through parameters, i.e., abstract and universally definable syntactic polymorphisms, the PCM quantifies language differences/similarities across languages and even language families into a synthetic measure.

In this work, we want to find out if the apparent lack of gene-language correlation among FU-speaking populations may hide some deeper congruence determined at possibly different stages by the relationships of FU with its IE and AL neighbors and by their migration history. To do so, we will estimate the degree of similarity between languages from the three families using the PCM.

The amount of secondary contact between individual pairs of languages can in principle be measured from lexical comparison. Indeed, lexical borrowings between Indo-European and Uralic [25,26,27] have been used to show historical contact between the two families. In principle, by counting for each pair of languages how many loanwords have been exchanged in either direction and reducing them to a single figure, one could estimate the amount of borrowing and use it as a distance measure to be compared with genomic data. However, presently, no general database summing up this kind of information in an exhaustive and uniform fashion exists, nor an algorithm for non-arbitrarily performing such calculations.

In this respect, complementing traditional lexical insights with comparison of syntactic parameters presents some particular advantages: a dataset for many relevant languages is already available [24], its historical information has been evaluated, and a syntactic distance is readily computable for every pairwise relationship within the whole set of languages used; such a distance sums up both potential vertical and horizontal signals into a single figure, paralleling well the genetic distances among the corresponding populations, which can be influenced by both splits and successive admixture.

In the course of the analysis, some individual taxonomic insights from classical linguistic levels (Ref. [28,29] and the other references mentioned above), will also be resorted to and compared with the syntactic signal, whenever informative.

The conclusions we reach about the geographical and demographical history of FU will have some indirect consequences on the current debate about the prehistory of IE speakers. As for the latter, let us recall that, despite a long tradition of studies, it is still debated whether early IE languages came into Europe from the Pontic-Caspian steppes (and spread West in the Bronze Age [30,31]) or from Anatolia (and spread with the dispersal of early Neolithic farmers [16,32,33]). Thus, we compared the syntax of several AL- FU- and IE-speaking populations with the available genome-wide data, both contemporary and ancient, in the area of interest. Of particular relevance was one Bronze-Age population from the Pontic-Caspian steppe, the Yamnaya, the likely source of the Bronze-Age migration leading to a Westwards diffusion of DNA of Central Asian origin and, according to some authors, of IE languages in Europe [34,35,36,37]. By contrast, a recent analysis of Asian genomes suggested that the spread of IE languages in South Asia and Anatolia may have little, if anything, to do, with migration from the Pontic-Caspian steppes [38]. An analogous uncertainty surrounds the homeland of early FU speakers, whether in the river Volga basin or further East, in Siberia [39].

Our multidisciplinary approach comparing grammars and genomes will ultimately help us better frame the evolution of this cultural and biological diversity in western/Central Eurasia, reinforcing the idea of widespread congruence between the two types of variables.

## 2. Materials and Methods

### 2.1. Genomic Dataset

The dataset analyzed in this study comprises the high-coverage sequenced genomes of 45 individuals from 17 populations from Eurasia (Appendix A). The samples were collected from Pagani et al. (2016) [40] and downloaded from the public database ENA (European Nucleotide Archive). For the sake of equal representation, a random subset of three individuals per population was chosen for populations with a larger sample size, to perform all the analyses.

Ancient and modern Genome-wide SNP array data from Ref. [41] were used to estimate Outgroup *f3*-statistic and *qpAdm* analysis (Appendix A, respectively).

### 2.2. Dataset Preparation

Samples from Ref. [40] were in Complete Genomics MasterVar format files (reads mapped against the human genome reference hg19/GRCh37). The MasterVar file was converted into a Variant Call Format (VCF) by the cgatool mkvf (version 1.8.0.1) from Complete Genomics. The VCF file created only contains SNP variants with a high confidence (>40 dB). All the VCF files from the different individuals were merged using BCFtools (version v1.6-36) merged with the option “-m none” to output the multiallelic sites in different lines. All duplicated variants were excluded from the data. The VCF files were phased using SHAPEIT2 (version v2.r837) using the 1000 Genomes phase 3 haplotypes as a reference panel, as recommended. Heterozygous sites not present in the 1000 Genomes data were left unphased. In the end, genotypes were obtained for 11,931,455 autosomal SNPs.

### 2.3. Principal Component Analysis

A general description of genetic variation was obtained by Principal Component Analysis (PCA). QTLtools [42] (version v1.1) was used on scaled and centered genotype data on relatively independent (50 Kb distance) and non-rare variants (minor allele frequency = 0.05).

### 2.4. Genomic Distances

Weir and Cockerham’s genomic distances between populations were calculated by the 4P software [43] (version 1.0). Genomic regions that may be under selection were masked using bedtools subtract (version v2.26) and variants with a missing call rate exceeding 10% were excluded, resulting in a total of 9,881,752 autosomal SNPs.

### 2.5. Linguistic Dataset

For the analysis of linguistic data by PCM, we used the 94 binary parameters and their settings defining properties of nominal structures for 69 modern Eurasian languages recently employed in [24] and accessible from https://github.com/AndreaCeolin/FormalSyntax. Theoretical background and more technical details about the structure of the parameter system and the settings of the individual values are found in [44,45,46].

The original dataset of 69 languages has been reduced to a subset of 34 IE, FU and AL languages, to improve resolution on the 17 populations for which genetic data are available and their neighbors.

Data were available for the main FU subfamilies (Ugric, Volgaic, Permic, Balto-Finnic), with the exception of Lapp (Saami). For three languages, Mari (Cheremiss), Udmurt (Votyak) and Khanty (Ostyak), we encoded two diastratic variants; the two Even (Tungusic, AL) varieties are instead diatopic. For details, see Ref. [24]. The relevant IE languages belong to three subfamilies, namely, Indo-Iranian, Germanic and Slavic (see Appendix A).

### 2.6. Linguistic Distances and Phylogenies

A matrix of pairwise syntactic distances was derived from the data matrix for the 34 relevant languages, by means of a Jaccard–Tanimoto formula (see [24]) reported below:Δ Jaccard (A,B) = [N_−+_ + N_+−_]/[N_−+_ + N_+−_ + N_++_]
where A and B are languages, Nxy indicates the number of positions where the string A has value X and B has value Y. The binary strings are interpreted as indicative of the presence (+) or absence (−) of traits (one per position in the string).

The distances inferred are summarized in Appendix A and visualized in a heatmap (Appendix A). By means of a Principal Coordinate Analysis (PCoA, also called MDS, Multidimensional Scaling), calculated using the software PAST [47], we visualized the syntactic relationships between languages (Figure 1). We also represented the syntactic data in tree form through a UPGMA tree (Figure 2) using the bootstrapping procedure described in [24], in combination with the software PHYLIP [48] and Mesquite [49], and through a character-based Bayesian tree, using BEAST v1.83 [50] (Appendix A). 

### 2.7. ChromoPainter and fineSTRUCTURE

ChromoPainter [51] (version v2) is a method to quantify distances between individual genomes. This method uses sampled chromosomes as “donors” and matches (or “paint”) other chromosomes to the donors’ DNA, thus quantifying similarities among individuals based on shared blocks of SNPs. In the heatmap, each square represents the number of DNA segments that each row (recipient) copies from each column (donor).

We used ChromoPainter output to cluster individuals into genetically homogeneous groups using fineSTRUCTURE [51] (version 2.1.3), a powerful approach for inference of fine-scale population structure from haplotype data. Each individual is presented as a matrix of non-recombining genomic chunks received from a set of multiple donors. Clusters of individuals are then inferred from the patterns of similarities among these copying matrices, by a Bayesian approach, and the tree is finally plotted using FigTree (version 1.4.2).

### 2.8. Outgroup f3-Statistics

We performed an *f3* analysis using the *qp3Pop* package in ADMIXTOOLS (version 412). The outgroup *f3*-statistic (X, Y; Outgroup) is a function of shared branch length between two genomes, say X and Y, in the absence of admixture with the outgroup. Y is extracted from a set of individuals, among whom we look for the most closely related to the individual under exam (X). Throughout the analysis we used the African Yoruba as an outgroup that we assumed to diverge from population X before all the other populations were analyzed. In this analysis, high values of *f3* indicate that X and Y are genetically closer.

The modern samples from Pagani et al. (2016) [40] used in this study were merged with the Yamnaya, Anatolian, Sintashta and Nganasan individuals from Ref. [38] and used as source populations. Variants with a missing call rate exceeding 10% were excluded, resulting in 249,286 SNPs suitable for the analysis.

### 2.9. Modelling Admixture

Using the *qpAdm* package in ADMIXTOOLS (version 412), we estimated the proportions of ancestry in a *Test* population deriving from a mixture of three reference populations by leveraging shared genetic drift with a set of outgroup populations. The reference populations used were: Yamnaya, Anatolia and Nganasan (used here as a proxy for the genetically still undescribed Siberian population). As outgroup populations, we used: Han, Mbuti, Karitiana, Ulchi and Mixe. The detail: YES parameter, was set, which reports a normally distributed Z-score for the goodness of fit of the model (estimated with a Block Jackknife).

## 3. Results

### 3.1. Linguistic Analyses

#### Syntactic Comparison

The PCoA inferred from syntactic data (Figure 1) shows a first, neat division between the IE languages, with positive values of the first component (accounting for 78% of variation), and FU and AL, all found in the left area of the graph. In that area, the second PC (accounting for 11% of variation) separates FU from AL. In sum, each group appears to form a well-defined cluster. While the clouds corresponding to IE and AL are compact, although with individual outliers (Indo-Iranian and Buryat, respectively), the FU languages appear more scattered. Finnish and even more so Estonian fall particularly close to the IE languages. Such a resemblance between the Balto-Finnic group of FU and IE emerges even more neatly in the Bayesian phylogenetic analysis (Appendix A), where the Balto-Finnic node joins the IE cluster rather than the FU one. The second important aspect that emerges from the PCoA is a split between IE and the other two groups, which might in turn hint at some closer FU–AL relatedness.

In the UPGMA tree (Figure 2), languages from the same family, IE, FU and AL, neatly cluster together without exception; FU languages form a monophyletic cluster within which the Balto-Finnic (Finnish and Estonian) and Ugric (Hungarian and Khanty) families are well identified, with the addition of a further node comprising geographically closer Udmurt (Permic) and Mari (Volgaic) [29,52]. The latter node occurs closer to Ugric than to Balto-Finnic, in disagreement with some traditional, though not uncontroversial [52], classifications.

The outlying positions of Balto-Finnic (Finnish and Estonians), Indo-Iranian (Pashto, Marathi and Hindi) and Mongol (Buryat) within the three groups are also visualized in the Heatmap of the syntactic distances (Appendix A).

### 3.2. Genetic Comparison

#### 3.2.1. Population Structuring in Eurasia

We selected 17 populations—seven speaking IE, six FU and four AL languages—for which whole-genome data were available (Figure 3a; Appendix A). The first principal component (Figure 3b) mostly reflects geography and separates eastern from western Eurasian populations, whereas the second component separates western Eurasians along a north–south cline. The AL-speaking populations fall into a single cluster along the first PC axis. The European IE-speaking populations form a cluster along the PC2 axis, separated from the Iranians, the latter belonging to the Asian group of IE languages.

Conversely, the FU-speaking populations are scattered along the PC1 axis. Estonians fall within the IE diversity at the negative end of the *X*-axis, while Finns occupy an intermediate position between the IE speakers and the FU-speaking Udmurt and Mari people, i.e., the modern inhabitants of the Pontic steppes (Figure 3b).

#### 3.2.2. Genetic Distances between Populations

Next, we calculated genetic distances (*Fst*) between pairs of populations (Figure 4). All AL and IE speaking populations are genetically closer to other populations of their language family than to populations belonging to a different family. Instead, that is not the case for the FU speakers; all of Estonians, Finns and Hungarians are genetically closer to their respective European neighbors speaking IE. In addition, among the eastern populations, the Mari and Udmurt seem genetically more similar to the other Europeans than to the AL speakers. Exceptions are the easternmost and Trans-Uralic Khanty (Ostyaks), which seem equally close to Mari, Udmurt and most of the AL speakers. This observation can be reconciled with historical data, which place the origins of the Khanty people in the Russian steppes followed by a northward migration into western Siberia in about 500 AD (500 BCE) [53].

#### 3.2.3. Shared Haplotypes

In the analysis of genetic distances, each single-nucleotide polymorphism is independently considered, regardless of its association with other polymorphisms. To analyze the patterns of population resemblance in finer detail, we thus moved to the haplotype level, using ChromoPainter and fineSTRUCTURE (Figure 5). This approach does not depend on prior information on sample groupings and operates instead with data-driven natural groups defined by patterns of haplotype sharing.

This approach also led us to identify three main genetic groups, broadly corresponding to the three main language families. However, as already observed in the *Fst* analysis, there were exceptions. The western FU-speaking populations (Estonians, Finns and Hungarians) seem to mainly share co-ancestry with the other Europeans, regardless of the language spoken. Conversely, among the eastern FU speakers, Udmurt, Mari and Khanty, there is a high level of haplotype sharing. In addition, this analysis revealed for the first time some co-ancestry of Finns (and partly Estonians’ and Hungarians’) with AL speakers of Siberian origin.

The evolutionary tree inferred from these data (fineSTRUCTURE cluster analysis; Figure 5b) shows two deep splits, the first isolating all AL speakers, and the second separating eastern FU speakers from a group composed by western FU and IE speakers. All this could even point to different ancestries for the FU-speaking populations, with phenomena of horizontal language diffusion leading them to a shared linguistic identity. However, lexical analyses and, in a more modulated fashion, even the syntactic ones support an original FU linguistic unity, later fragmented by northward and westward migrations and contacts. To better understand these results, we resorted to ancient DNA.

#### 3.2.4. Affinities between Modern and Ancient Populations

Our genetic analysis showed the Udmurt and Mari to be closer than the Khanty to European populations (Figure 4 and Figure 5a). We hypothesize that this observation may be related to shared ancestry with Yamnaya, an ancient pastoralist population that lived in the current Udmurt and Mari territories, around the Pontic-Caspian steppes, and that expanded into Central and western Europe in the third millennium BCE, contributing a Caucasian genomic component that nowadays is widespread in Europeans [35,37]. We tested for genetic continuity from the ancient Steppe populations, Yamnaya (~4700 yBP) and the more recent Sintashta (~3900 yBP) on the one hand [35,36], to current Udmurt and Mari on the other. An ancient Anatolian sample [54] was also included in our tests, potentially accounting for the genetic legacy of early farmers from the Near East.

We formulated outgroup *f3*-statistics of the form *f*3(AP, MP; Yoruba), where *AP* was represented in turn by each of the three ancient populations, and MP was each of the modern samples in our dataset (Figure 6 and Appendix A). In general, we found all ancient samples to share more genetic drift with modern Europeans and Russians than with non-European populations. Among the eastern populations, the Udmurt and Mari are the ones sharing the most genetic drift with Yamnaya and Sintashta; on the other hand, the Iranians (IE) are the Asian sample closest to the Anatolian farmers, in agreement with recent findings [37]. In addition, within the European populations, the *f*3 values show opposite trends for the Anatolian and the Yamnaya/Sintashta, the former sharing more genetic drift with southern and Central Europeans (Croats and Germans) and the latter being closer to Northeast Europeans, including the FU-speaking Estonians and Finns, once again in general agreement with previous findings (e.g., Ref. [35]). It is interesting to notice the peculiar behavior of the Hungarians. They appear much closer to the ancient Anatolians than to the Yamnaya, which is common among southern European populations; however, they are the modern Europeans sharing most genetic drift with the Sintashta. This may be indicative of a relatively more recent genetic contact between them and the Steppe populations, i.e., after the process leading to the spread of the Yamnaya component into Europe.

Contrary to what could be expected, the modern FU inhabitants of the Russian steppes, Mari and Udmurt, appear more distant from Yamnaya than Estonians and Finns. One possible explanation would be the presence, in their genomes, of a Siberian-related component, known to be widespread in contemporary Central and North Asian populations [55,56,57,58]. We tested for its presence in our samples by modelling Nganasan, a population of residual speakers of a moribund Samoyed language (i.e., distantly related to FU) from the Taymyr Peninsula, as a proxy of the carriers of this Siberian component (as also in Refs. [33,38]). We did find support for the presence of such a Siberian component among Mari and Udmurt; the outgroup *f*3 statistics of the form (Nganasan, MP/AP; Yoruba) showed that Udmurt and Mari are indeed closer to Nganasan than Yamnaya, which shared similar *f3* values with other European population with regards to Nganasan. Figure 6b shows a clear trend; the Nganasans share more genetic drift with all AL speakers, followed by Udmurt and Mari, and then by European populations, no matter if FU- or IE speakers.

To further test whether the peculiar genetic position of the Udmurt and Mari is really associated with the higher presence of a Siberian genetic component in their genome, we ran a *qpAdm* analysis (Figure 7 and Appendix A). All the FU-speaking populations were successfully modelled as a mixture of Yamnaya, Anatolian and Nganasan-related ancestry, with the exception of the Khanty, who seem to have no Anatolian ancestry. In particular, the Mari and Udmurt genomes appear to contain a large component that can be related with a Siberian genetic ancestry, confirming our expectations. Furthermore, this Siberian ancestry is present, at low though non-negligible percentages, in the western FU-speaking Finns (but less saliently in Estonians).

## 4. Discussion

### 4.1. Syntactic Diversity

Syntax distinguishes IE, FU and AL languages quite well, although IE and AL have single outliers (Indo-Iranian and Buryat, respectively). Conversely, the FU family turns out to be less compact, in spite of the greater geographic spanning and population size of IE, and of the weaker lexical evidence purportedly supporting AL (see Ref. [12] for the state of the debate). The whole family appears scattered and in some structural contiguity with their eastern and western neighbors.

A previous study comparing Uralic lexical data, including the FU speakers, had suggested that some secondary contact played a role in the divergence of these languages [59]. The scattered pattern is now more clearly observed and measurable through the cross-family application of our syntactic analysis. The outlying position of Estonian and Finnish among FU languages is evident (especially see Figure 1 and Appendix A). As for the other western FU language, Hungarian, qualitative analysis shows that the language shares some parameter values with IE, as opposed to the rest of FU. Yet, such similarities do not emerge in the trees, possibly reflecting the much later arrival in Europe of the Hungarian language [60].

The very distribution of similarities and differences in the syntactic parameters suggests that the scattering of FU languages is likely to be secondary, i.e., due to cultural contacts. Indeed, there is no evidence of potential convergence of Khanty, Udmurt, Mari with IE. In addition, the main syntactic changes detaching Finnish and especially Estonian on the one hand, and Hungarian on the other, from the other FU languages are that they are: A. different from each other; B. unidirectional, i.e., of a kind that is often acquired but hardly reversed; C. shared with neighboring IE languages at the time of the contact with the respective FU languages [24]. This tends to exclude that these properties might be ancestral (proto-Uralic) and lost in the more eastern varieties because of recent convergence with Asian languages. The similarities of eastern varieties with AL languages are, instead, more ambiguous as to whether they may be shared inheritance or a secondary effect.

In sum, our syntactic phylogenetic analysis supports the original wisdom that FU has been a monophyletic cluster, and is well compatible with the traditional view that the western FU languages have reached Europe from the East at some ancient point; but syntax also detects and measures the pattern of secondary similarities with neighboring languages.

### 4.2. Genome Diversity

The three main population groups identified by the linguistic analysis are also biologically differentiated; however, while IE and AL samples form distinct genetic clusters, both in the PCA and ChromoPainter analyses, a peculiar pattern emerges within the FU language family. While the Khanty show affinities with a well-differentiated cluster comprising all AL speakers, the other FU speakers appear to be part of a broad group, including all IE-speaking individuals. In particular, the western FU speakers, namely, Finns, Estonians and Hungarians, are genetically closer to IE populations in Europe than to the Asian UR-speaking populations. Estonians and Finns also share more ancestry with each other than with the Hungarians. This genetic similarity can reflect: (i) a different source of steppe ancestry in the Hungarians (more closely related with the Sintashta) than in Finns and Estonians (genetically closer to the Yamnaya) (Figure 6a); and/or (ii) a lower contribution of Siberian ancestors to the Hungarian genomes than to the Estonians and especially the Finns (Figure 6b).

### 4.3. Comparison of Genetic and Linguistic Results

Judging whether or not linguistic and genetic data mirror each other may be partly a matter of taste. However, there is little doubt that the syntactic and genomic findings of this study match and corroborate each other. In five out of six cases, linguistic and genetic evidence were consistent (Table 1), the only exception being the third one.

In this field, however, exceptions are as interesting as the rules, as they call our attention to phenomena that need be further investigated. By looking into the syntactic features of western FU languages, and into their speakers’ genomes, we could recognize peculiar processes affecting the demographic history of people speaking Estonian, Finnish and Hungarian.

Indeed, the Bayesian syntactic tree matches the strong similarity between IE and Balto-Finnic revealed by the genomic tree and PCA, but, on the whole, syntax supports the FU unity to a stronger extent than genetics, and neatly recognizes the Ugric group (Hungarian and Khanty). On the contrary, at the genetic level, the FU-speaking populations cluster according to geography, with Hungarian speakers close to Central Europeans, Khanty speakers close to their eastern AL-speaking neighbors, and the steppe-dwelling Mari and Udmurt speakers in an intermediate position. This result suggests that syntax can also capture secondary demographic events (e.g., population admixture), which genetics can identify only if they have entailed substantial demographic change.

In particular, syntax shows more limited secondary effects on Hungarian from its IE geographic neighbors and preserves well its historical similarity with Khanty. As we shall discuss later in this paper, historical data suggest that the establishment of Hungarian in Central Europe was the product of an episode of élite dominance, i.e., a deep change of language with limited demographic impact [4]. If so, the genomes of modern speakers of Hungarian were affected only marginally by the phenomena that radically replaced their language.

### 4.4. Demographic Scenarios: Linguistic and Genetic Evidence

The conclusions above, especially as summed up in the Synopsis of Table 1, reinforce our trust in the usefulness of gene/language comparison as a heuristic tool. We consider now what such approach tells us about the prehistory of a portion of western Eurasia.

While little is known about the ancient demographic history of AL populations, data are now available from pre-historic peoples of western Eurasia and the Near East. Analysis of genomes from pre-historic inhabitants of the Russian Steppes, the Yamnaya [36,37], identified a westward migration of people who contributed an ancestral component today widespread in Europe. Although no linguistic evidence was presented in that influential study, the authors linked the westward Yamnaya migration with the expansion of the IE languages. This hypothesis is based on two implicit but reasonable premises: one, that IE languages have been uninterruptedly attested in all the core of Europe for over two millennia now, and, two, that a high congruence (and consequent reciprocal predictability) of genetic and linguistic diversity is characteristic of Indo-European-speaking Europe (see [4], for example).

At the same time, other studies have related the presence of the Estonian and Finnish languages in Europe, instead, to a northward migration of people of ultimately Siberian ancestry [55,58].

Our multidisciplinary analysis seems to necessarily point to a more intricate scenario. Consider two points, again: first, a highest prevalence of the ancestral Yamnaya component is now attested precisely among Estonians and Finns (cf. Figure 7); second, the genetic/linguistic comparisons summarized in Table 1 have uncovered that, at a finer-grained analysis, a substantial gene/language congruence exists within the FU family.

Linguistically, it is clear that speakers of AL, IE and FU languages (along with their Uralic cousins, the Samoyeds) have long formed three separate groups, which had contacts leading to various degrees of linguistic and biological exchange. The data cannot exclude that some even more ancient unity—or close contact—may have involved at least two of the groups, proto-Uralic and proto-Altaic speakers [24,39]. Specifically, our syntactic database confirms the existence of a neat Ugric subfamily of FU, and detects remarkable, though measurably different, degrees of Indo-Europeanization for Finnish and Estonian.

Now, the classical and solid linguistic relatedness of Mari and Udmurt with the Balto-Finnic languages (partly obscured in the syntactic phylogenies because of the well-justified parametric interference of the latter with IE), as well as the genetic relationships of these modern populations (Mari and Udmurt in the steppes and Estonians and Finns in north-eastern Europe) with the ancient Yamnaya (Figure 6a and Figure 7), suggests a demographic and linguistic northward expansion of people with steppe-related ancestry into the Baltic area. This is consistent with the fact that Balto-Finnic preserves ancient Indo-Iranian (hence southern) IE loanwords more than other Uralic varieties and with the general hypothesis of an expansion of the FU languages from the Volga river basin [26,28,61,62].

A likely link of the expansion of FU languages into northeastern Europe with that of the Yamnaya (and not direct Siberian) genomic ancestry is of a chronological order. Some possible dates of the FU diversification, estimated between 5000 and 4000 yBP in [26,60,62], coincide in time with the first appearance of the Yamnaya genomic component in ancient European populations [35,36,41]. Particularly in the Baltic region, analysis of ancient DNA has dated the first contacts between Yamnaya migrants and the local communities around the early Bronze Age (5000–4500 yBP), involving Baltic hunter–gatherers with no Neolithic ancestry [63]. Such chronological evidence overlaps with the early dates of lexical borrowing between IEs and Balto-Finns mentioned above [25,26,27].

A reconstruction of the lexical history of the FU languages suggests a later northward expansion from the Baltic area to southern Finland, possibly around 2000–1600 yBP [60]. This expansion was accompanied by the separation of the northern (Finnish) from the southern (Estonian) group, which may have involved two successive linguistic steps [29]. It is this last diversification that overlaps in time with the first appearance of the Siberian genomic component in ancient remains around the Baltic area [58]. Therefore, ancient DNA and archaeological studies agree in suggesting that people related to the Yamnaya culture moved from the coastal Baltic areas into southern Finland around 2000–1600 yBP (i.e., much later than their linguistic presence around Baltic is first attested by their lexical exchanges with Slavic and Germanic tribes), where they first came into contact with the ancestors of modern Saami (Lapps) [57,58,60,64,65]. Traces of this contact, and of the limited admixture that must have followed, are still detectable in the genomes of Finns [66] (also in Figure 7) more than in the Estonian ones. The different degrees of syntactic similarity we were able to quantify in this study between Estonian and Finnish with respect to IE, on the one hand, and the other FU languages, on the other (Figure 1 and Appendix A), seem in agreement with this secondary migration model (although by themselves they could not be informative about the direction of such exchanges, whether South-to-North or North-to-South).

Finally, our genomic analysis is readily compatible with a second event of expansion of the FU languages (possibly through the Russian steppes) into Europe without involving Siberian mediation. That is the case of the FU speakers in Hungary. There is historical evidence that at the beginning of the Medieval era, the language spoken in nowadays Hungary was still Late Latin (at least as an official language), later subject to the effects of Slavic, Germanic and Avar invasions [67]. The main linguistic shift can be approximately dated around 895–905 AD, when people coming from the East conquered Hungary, imposing their own Ugric language [67,68]. Ancient DNA studies of the invaders have shown that they were genetically close to the Sintashta of the steppes, and apparently unrelated with Siberian ancestors [69], in fine agreement with our genetic analysis. Therefore, the presence of a FU language in Europe is not correlated with the presence of a Siberian component in the DNA of its speakers.

### 4.5. Speculations on the Diffusion of IE into Europe

Linguists and archaeologists have long discussed the timing and modes of spread of IE languages in Europe. Gimbutas [30] associated it with the westward spread of the Kurgan culture, from the Pontic steppes during the Bronze age, whereas Renfrew [33] saw it as a consequence of the Neolithic farmers’ demic diffusion from Anatolia (see also Refs. [16] and [70]). These alternatives (the Steppe and the Anatolian hypothesis, respectively) are paralleled at the genetic level, by studies supporting demic dispersal, respectively, of Yamnaya-related populations in the Early Bronze Age [35,36], and of Anatolia-related populations during the Neolithic transition [2,3,7,38,71,72].

The genomic similarity between the Yamnaya and the first FU speakers of Europe may be difficult to reconcile with the view that the Yamnaya were also the first who introduced IE languages in Europe, as suggested by studies of genomic data not supported by linguistic analyses [35,36]. One possibility, supported by a study of Iberia [73], is that the arrival in Europe of the Yamnaya genomic component did not necessarily entail the same linguistic changes in all areas. In the absence of adequate data to formally test this hypothesis, we still may speculate that the small, but non-negligible, ancestry component associated with the Anatolian Neolithic [38] among the Yamnaya may reflect the previous northward gene flow from the Near East into the Pontic steppes. If so, it would be possible to reconcile genetic evidence for the Neolithic demic diffusion from the Near East, linguistic evidence on a Near East center of IE diffusion [16,33,38,70,74], and data suggesting a role of Yamnaya people in spreading both IE [35,36,75] and FU (this study) languages by imagining the existence of some linguistic diversity within the Yamnaya-like populations and concluding that IE languages have entered Europe in two moments and by two routes. The first one would correspond to the main Neolithic expansion, Northwest into southern and then Central Europe, but also North, towards the Pontic Steppes. The linguistic impact of this migration would have not been the same for all people in the Pontic steppes; some would retain their original FU languages, and some would acquire an IE language. The former would then mostly move towards the Baltic Sea area, whereas the latter would correspond to the IE-speaking populations dispersing in Central Europe in the Bronze Age [36,76], giving rise to the Bell Beaker and Corded Ware cultures.

## 5. Conclusions

This study exemplifies how appropriate quantitative and qualitative tools allow one to measure cross-family language variation, offering a novel insight into human prehistory and generating testable hypothesis for large-scale genomic analyses. Of course, we must warn about the risk of over-interpreting correlations between languages and genes, especially in the absence of accurate dates of linguistic diversification and expansion, which are not yet well established. Furthermore, full inference of complex processes requires the study of broader datasets than available for the present study.

Nonetheless, on the whole, our analysis, taking advantage of various linguistic features, including the syntactic ones, which can be compared across families and are sufficiently stable in time, suggests that Darwin’s prediction of a general correspondence between biological evolution and language transmission is still generally valid, and that exceptions to this rule are both limited (more than it may appear from simply relying on traditional and non-quantitative language taxonomies, such as, e.g., in [68]) and extremely useful for a detailed reconstruction of human past.

## Figures and Tables

**Figure 1 genes-11-01491-f001:**
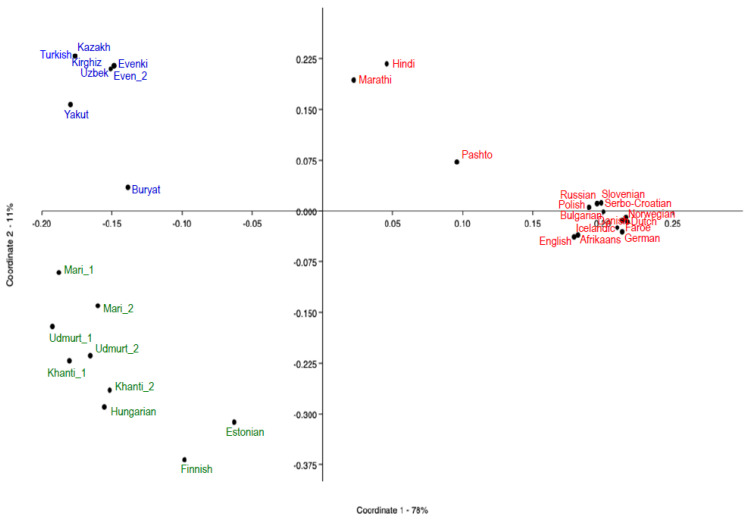
Principal Coordinate Analysis (PCoA) from the syntactic distances in 34 Eurasian populations. Language groups coded as follow: Finno-Ugric (**green**), Altaic (**blue**), Indo-European (**red**).

**Figure 2 genes-11-01491-f002:**
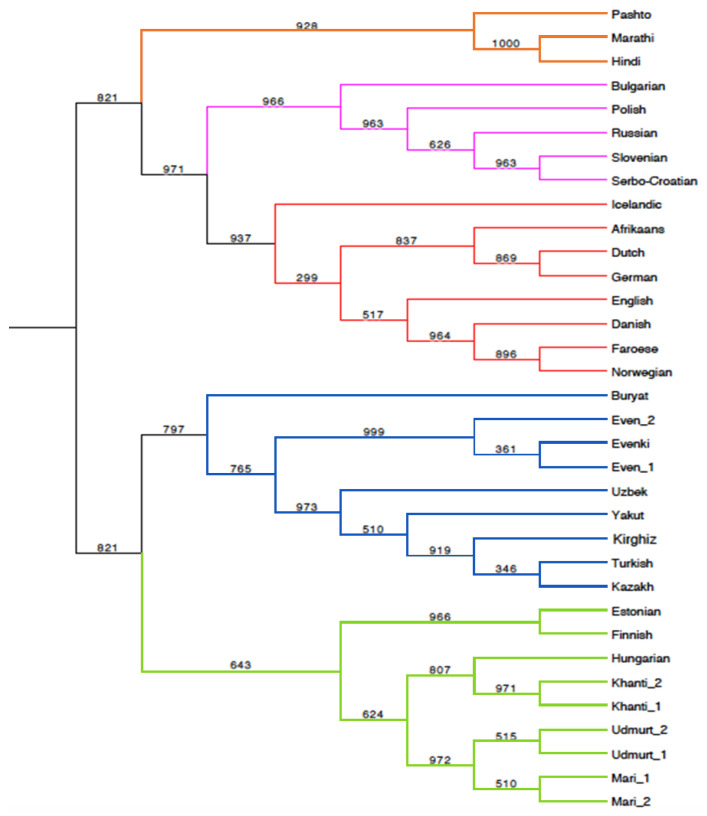
UPGMA tree inferred from the Jaccard syntactic distances. Bootstrap values, base = 1000, at the nodes. Orange = Indo-Iranian (IE—Indo-European), pink = Slavic IE, red = Germanic IE, Blue = (AL—Altaic), Green = (FU—Finno-Ugric)

**Figure 3 genes-11-01491-f003:**
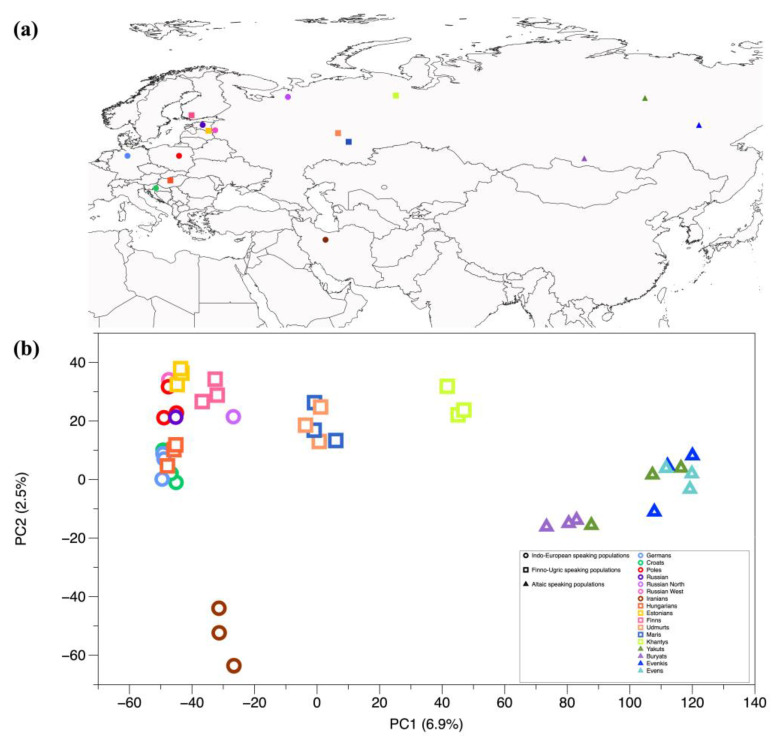
Geographical locations and Principal Component Analysis (PCA) of genomic variation. Populations speaking an IE, FU and AL language are represented by circles, squares and triangles, respectively. (**a**) Geographical locations of the samples in this study. (**b**) Projection on two dimensions of the main components (PCA) of genomic variation in IE, FU and AL speaking populations.

**Figure 4 genes-11-01491-f004:**
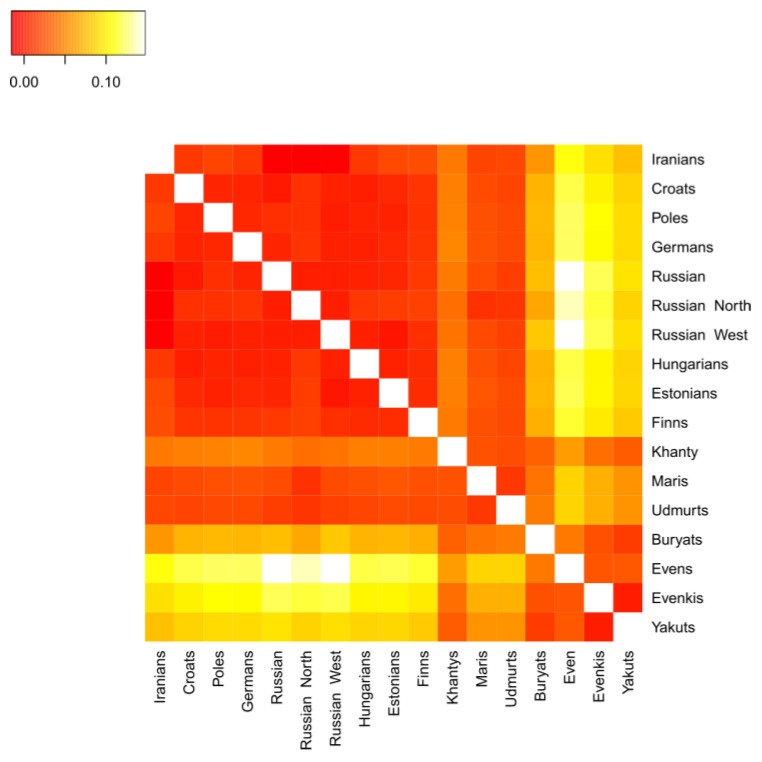
Pairwise genetic distances between Eurasian populations. Darker colors indicate that populations are genetically closer, whereas lighter colors indicate that populations are genetically distant.

**Figure 5 genes-11-01491-f005:**
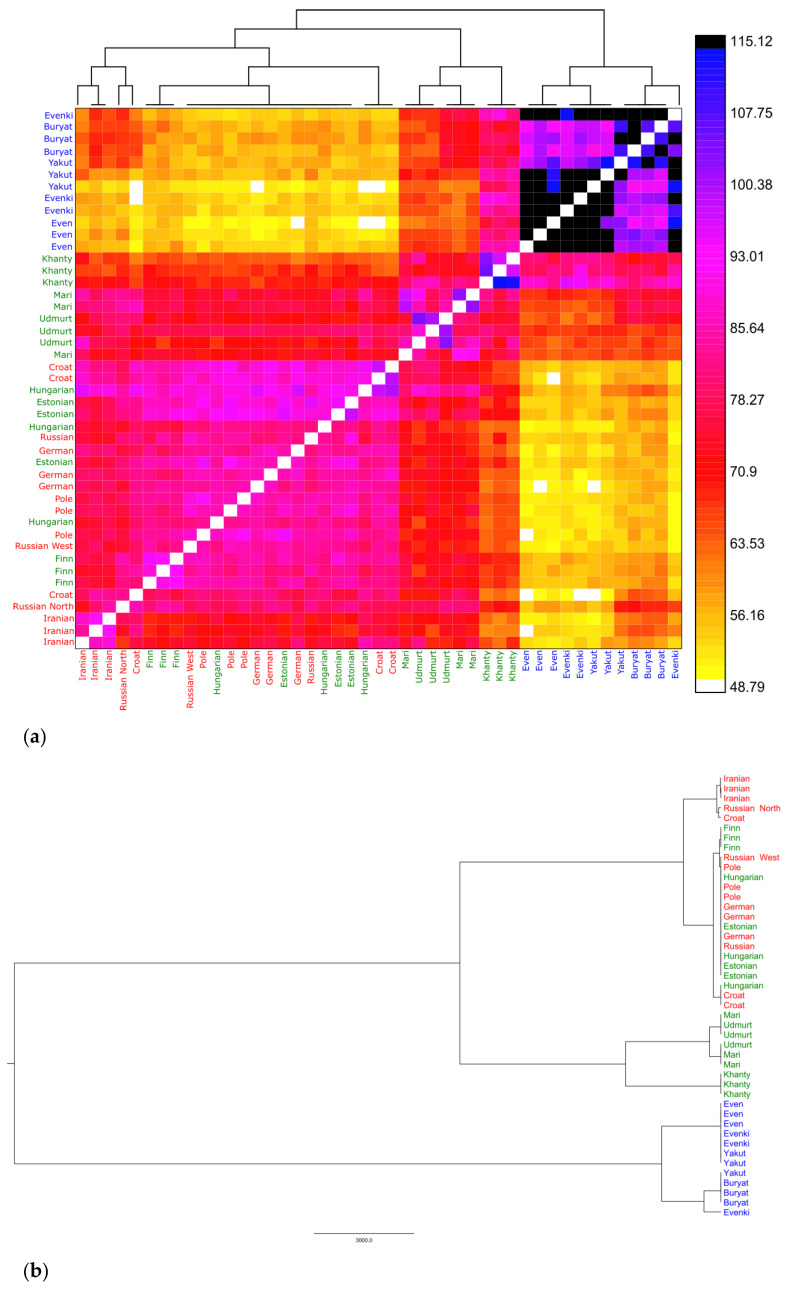
Estimates of shared ancestry between Eurasian individuals. (**a**) Co-ancestry heatmap. Each of the 51 individuals is represented as a row, where each pixel represents the level of co-ancestry (higher for darker colors) shared with each of the other individuals. (**b**) fineSTRUCTURE cluster analysis obtained from the co-ancestry matrix. Red = IE; Green = FU; Blue = AL.

**Figure 6 genes-11-01491-f006:**
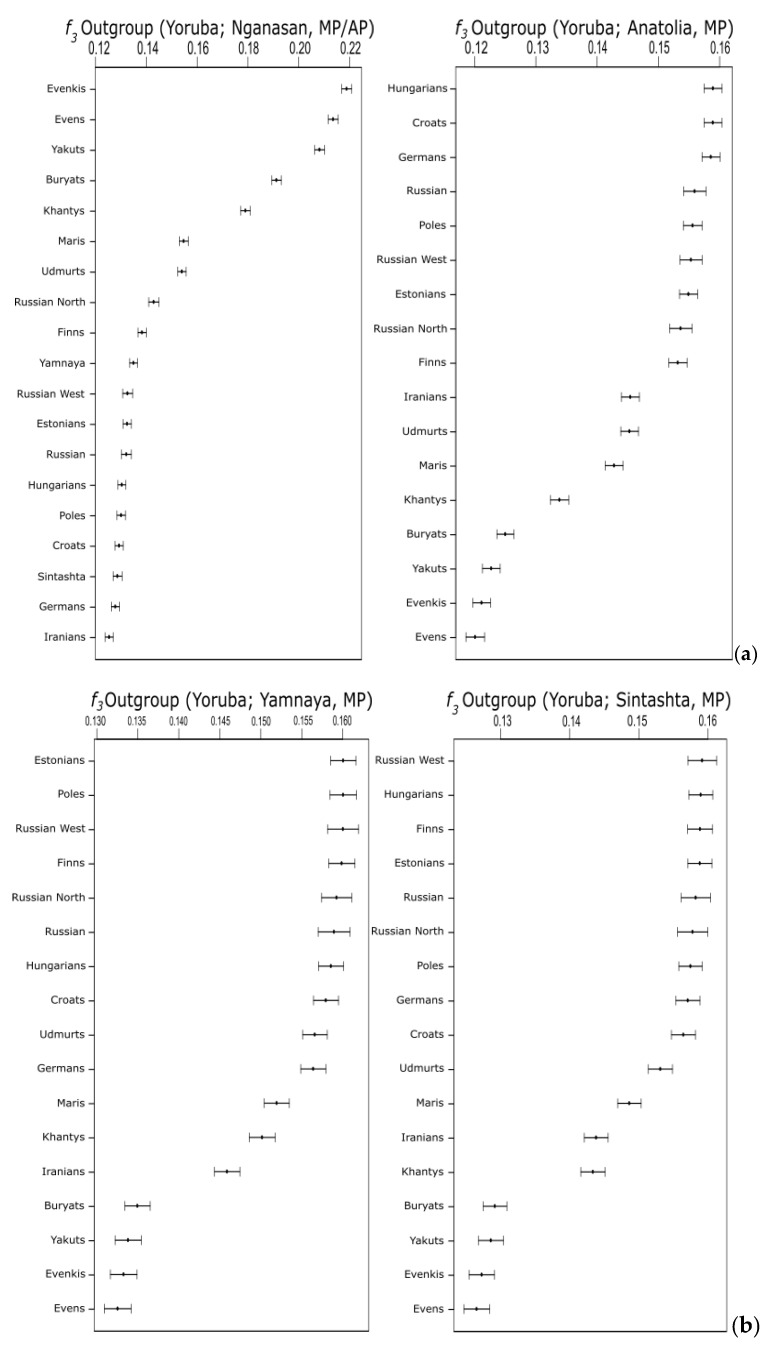
Outgroup *f3*-statistic analysis. Shared genetic drift between ancient and modern (MP) populations. (**a**) Shared genetic drift between Anatolian, Yamnaya, Sintashta, (**b**) Nganasan and modern/ancient populations.

**Figure 7 genes-11-01491-f007:**
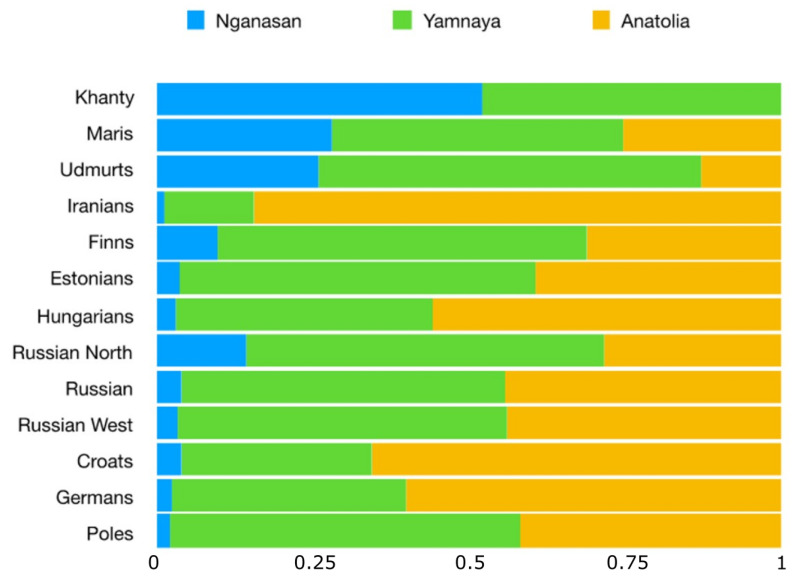
Admixture proportions from three sources estimated using *qpAdm.* Sources used were Nganasan, Yamnaya and Anatolia (percentages and chi-square values are shown in the Appendix A).

**Table 1 genes-11-01491-t001:** Synopsis of the main results of this study.

	Syntax	Modern Genomes	Ancient Genomes
1	AL languages form a cluster	AL speakers form a cluster	Higher Siberian component in AL speakers than in all the other populations
2	Indo-Iranian languages distinct from European IE languages	Indo-Iranian speakers distinct from other IE speakers	Higher Anatolian component in Indo-Iranian speakers than in other IE speakers
3	FU languages separated from IE and AL	In the tree, FU speakers and IE speakers fall in the same cluster	Yamnaya and Anatolian components similar in western FU speakers and their European IE-speaking neighbors
4	Estonian closer to IE and more distant than Finnish from other FU languages	Estonians closer to IE speakers than Finns	Siberian component lower in Estonians than in Finns
5	Mari, Khanty and Udmurt closer to AL than to IE languages	Mari, Khanty and Udmurt speakers more distant from IE speakers than Finns, Estonians and Hungarians	Higher Siberian component in Mari, Khanty and Udmurt speakers than in any other FU population
6	Easternmost FU Khanty least distant from easternmost Yakut of all AL languages	Khanty speakers halfway between the Mari/Udmurt speakers and eastern AL populations	Khanty speakers have the Siberian and Yamnaya component, but no Anatolian one

Note that ancient Siberian ancestry is (here and elsewhere: Refs. [55,58]) approximated by a modern population, Nganasans.

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
