# Peer review of "More Rule than Exception: Parallel Evidence of Ancient Migrations in Grammars and Genomes of Finno-Ugric Speakers"

_genes, 2020, doi:10.3390/genes11121491_

Round 1

Reviewer 1 Report

This paper investigates the complex relation of genes and languages among Finno-Ugric (FU) speakers , compared to their Indo-European (IE) and Altaic (AL) neighbors. The authors analyze genome-wide data in both ancient and contemporary populations, and note matching patterns: north-western FU speakers linguistically and genetically closer to their IE speaking neighbors, and eastern FU speakers to AL-speakers. The paper identifies the Pontic-Caspian steppes as the possible origin of the demographic processes that led to the expansion of the FU into Europe.

The paper has a clear structure, presents interesting results, and certainly merits publication. There are, however, some issues that in my opinion need to be addressed. First, as reference 22 is not yet available, it is rather difficult for readers to appreciate how linguistic distance was calculated.

My second concern relates to the binary parameters: since the authors only consider properties of nominal structure, some comments are in order. Why only nominal structure and do we expect a more complex/different picture if verbal structure is included? It would help if the appendix included the features the authors consulted and if an example of the central closeness could be given.

The authors probably know that they are currently several views on the nature of parameters, see e.g. Biberauer & Roberts (2012, https://www.mmll.cam.ac.uk/files/copil_6_9_biberauerroberts.pdf) who put forth a finer grained classification. How do the features the authors have in mind fare with respect to that?

Reviewer 2 Report

The paper is an interesting and innovative study on an important and actual topic. It attempts to approach the prehistory of the Finno-Ugric languages through multidisciplinary approach that combines results of linguistics and genetics. However, the results are only partially convincing, and some revisions are needed before the paper can be accepted for publication.

As a linguist, I cannot judge whether the genetic data and the results based on it are correct, but they are presented in a clear way that is understandable also to non-specialists of genetics. The authors argue that the genetic evidence supports the wide-spread view that Proto-Finno-Ugric originated around the Volga basin: it means that the genetic results agree with the consensus of linguistic prehistory, and I am certain that the results presented here will be useful and interesting for researchers of the history of the Finno-Ugric languages.

However, as a linguist I do not completely agree with the methods the authors use, as they approach the linguistic prehistory through exclusively syntactic data. I agree that historical syntax is important, but in the research on linguistic divergence, also phonology, morphology, and to a lesser extent lexicon, play an important role, but this is not reflected in the text of the paper.

The authors do mention that the novel model that they use, Parametric Comparison Method, is more advanced method than the comparison of vocabulary items. However, it has to be said that this method has not become so widely accepted in Finno-Ugric linguistics or in linguistic studies in general, that one could simply argue that this is the most reliable method. More discussion on this method would be welcome, as that would make the issue more transparent.

Research on syntax is without a doubt a useful way to approach this issue, but the authors should make it more explicit why they assume their new method is better than the other aspects of historical-comparative linguistics, namely phonology and morphology, that have played key role in linguistic classification.

This is my main criticism, as I am sure that now the issue remains rather unclear for even a linguist, not to speak of a reader who is familiar with genetics but not with syntax or other areas of linguistics.

I also do not completely understand how can the measures between Altaic, Indo-Europan and Finno-Ugric measured through the use of syntactic data: since these are three unrelated language families (and the status of Altaic as a family is even questioned), I understand that the syntactic data is useful in tracing contact-induced similarities, but I have trouble in understanding what this tells of the possible linguistic relatedness of these three families. Regarding the conclusions of the paper, I understand that the authors have observed similarities in the syntax of various Finno-Ugric, Indo-European and Altaic languages, and this is valuable information. However, I do not understand whether this should point to the conclusion that the three families are related to each other. This should be expressed more clearly.

It is well-known that the Altaic, Indo-European and Finno-Ugric languages have been in lexical contact, with many words borrowed into Finno-Ugric from various Altaic and Indo-European languages. Such loanwords have frequently been used in the discussion of the location of proto-languages and the dates of their divergence (see eg, Koivulehto 2001, and Kallio 2006; the latter is even included in the references of the paper). It would improve the paper greatly if the authors added a short discussion about how their conclusions based on syntactic data fit with the results of loanword studies.

The remarks above are not meant to be overtly critical: in the “Conclusions”, the authors are aware that research on complex processes requires larger datasets, and that the correlation of languages and genes should not be overinterpreted. If the authors added more details on traditional methodology of linguistic taxonomy to their discussion, it would be more transparent to non-linguistic readers that the research on linguistic prehistory and language diversification is a very complex issue that can be approached from various viewpoints.

In addition, there are small shortcomings and methodological issues that I would like to point out:

Lines 207–211: When discussing the syntactic distances between Finnish, Estonian and Udmurt, it would be useful to mention that shared innovations on other levels of language (phonology, lexicon) have also been suggested, and it has often been assumed that the Finnic branch, together with Mordvin, Mari and Permic (the branch that includes Udmurt) are mose closely related to each other than to the Ugric branch that includes Hungarian, Khanty and Mansi. For the sake of research history, this fact should be mentioned (although it is debatable whether phonological or lexical innovations indeed support this traditional view). Discussion of this issue can be found in Salminen (2002).

Also, note that Europe also includes a large part of Russia, and it is misleading to speak of Finnish, Estonian and Hungarian as all three Uralic languages in Europe. Mari and Udmurt are spoken in Europe as well. Speakers of Mari and Udmurt are called “Asian populations” by the authors (lines 292–293), but in reality the areas where Mari and Udmurt are spoken are located in the European side of the Russian Federation.

 472–474: It remains unclear to me why “Siberian influx seems to be too recent to explain the presence of the FU in the Baltic area”¨. Mentioning the arguments in the text would make this claim more transparent.

488: The authors mention the “Finno-Volgaic branch”. However, such a branch is not universally accepted by researchers of Finno-Ugric linguistics, and the authors should be cautious in assuming a date for the split of this branch, the existence of which is disputed. Salminen (2002) provides good arguments against a Finno-Volgaic branch.

492–493: Here a reference to Kallio (2014) should be added, as this is the most up-to-date discussion of the linguistic diversification of the Finnic branch, to which both Estonian and Finnic belong to. It is notable that South Estonian Võro-seto and Livonian are the earliest ones to branch off this, and the difference between Estonian (= Standard North Estonian) and Finnish is not the oldest split within the Finnic branch.

References:

Kallio, Petri 2014: Diversification of Proto-Finnic. – Joonas Ahola, Frog, Clive Tolley (eds.), Fibula, Fabula, Fact – the Viking Age in Finland. Studia Fennica Historica 18; Helsinki: Suomalaisen Kirjallisuuden Seura. 155–168.

Koivulehto, Jorma 2001: The earliest contacts between Indo-European and Uralic speakers in the light of lexical loans. – Christian Carpelan, Asko Parpola & Petteri Koskikallio (eds.), Early Contacts between Uralic and Indo-European: linguistic and archaeological considerations. Mémoires de la Société Finno-Ougrienne 242. Helsinki: Société Finno-Ougrienne. 235–264.

Salminen, Tapani 2002: Problems in the taxonomy of the Uralic languages in the light of modern comparative studies. – Лингвистический беспредел: сборник статей к 70-летию А. И. Кузнецовой. Москва: Издательство Московского университета. 44–55.
